# Hydraulic Characteristics of Lateral Deflectors with Different Geometries in Gentle-Slope Free-Surface Tunnels

**Jinrong Da [1], Junxing Wang [1,\*], Zongshi Dong [1] and Shuaiqun Du [2]**

1   State Key Laboratory of Water Resources and Hydropower Engineering Science, Wuhan University, Wuhan 430072, China
2   Power China Guiyang Engineering Corporation Limited, Guiyang 550081, China
\*   Correspondence: jxwang@whu.edu.cn; Tel.: +86-137-0718-2138

**Abstract:** The gentle-slope tunnel has been adopted in many high dams, and aerators are usually required for high operating heads. For such tunnels, the lateral deflector is superior to the traditional bottom aerator, which loses its efficiency due to cavity blockage and fails to aerate the sidewalls. However, unfavorable flow patterns such as water-wings and shock waves are induced by the lateral deflectors. To address this problem, two novel lateral deflectors are proposed, and their hydraulic characteristics are comparatively investigated together with the triangular deflector by means of model test and numerical simulation. The triangular deflector was revealed to form a wide cavity that allows for the free rise up of the water-wings inside the cavity, leading to the development of a buddle-type shock wave, whereas the two-arc deflector yields a jet with a fluctuating surface, which induces water-wings and further develops into diamond-type shock waves. In contrast, the cavity formed behind the two-arc deflector with a straight downstream guiding line is stabler and shorter, thereby restricting the development of the rising flow and preventing the formation of water-wings and shock waves. Moreover, the two-arc deflector with a straight guiding line exhibits higher energy dissipation capacities and thus is recommended in practical engineering design.

**Keywords:** lateral deflectors; gentle-slope tunnel; water-wing; shock wave; energy dissipation

## 1. Introduction

Due to complex topographic and geological conditions [1] as well as material transport difficulties, high dams usually adopt arch concrete or local material types [2,3], the dam-body flood-discharge capacity of which is significantly limited [4]. Consequently, complementary flood release structures are of great importance for a sound design of the entire project [4]. In the past century, the tunnel spillway has stood out from other alternatives [4,5], such as chute spillways and shaft spillways owing to its high flood-release capacity, good terrain compatibility, and decent construction cost [6]. To avoid potential cavitation and denudation damage [1,4,5,7], the flow pattern and velocity in the spillway tunnel have to be strictly controlled. The ideal flow pattern should at least meet the following two requirements: (i) the flow is free-surface flow without shock waves and with adequate space between the aerated surface and the tunnel soffit [8]; (ii) the flow velocity is within 25 m/s [8] and the structure surface is carefully smoothed [8,9]. Under such conditions, technically no aerator is required. In the past decades, the sagging dragon tail tunnel has been proven to effectively fulfill the above requirement and thus has gained popularity in many high dams (e.g., Jinping, Baihetan, Xiaowan [5], Wudongde [10], Shuangjiangkou [11]) in China. The underlying reason for this kind of tunnel's satisfactory hydraulic behavior lies in its longitudinal layout, which is reflected in its name 'sagging dragon tail'. It comprises an overflow ogee weir (dragon head) close to the reservoir, a long gentle-slope free-surface tunnel in the middle (dragon body) and a short open and steep chute connected to a ski-jump bucket (the sagging dragon tail) at the end, as shown

in Figure 1. With such a design, the flow acceleration is negligible inside the gentle-slope tunnel [4,5] and the high-velocity flow only occurs at the short steep chute, where the flow is usually highly aerated and the potential local structure destructions can be easily found and repaired once it occurs.

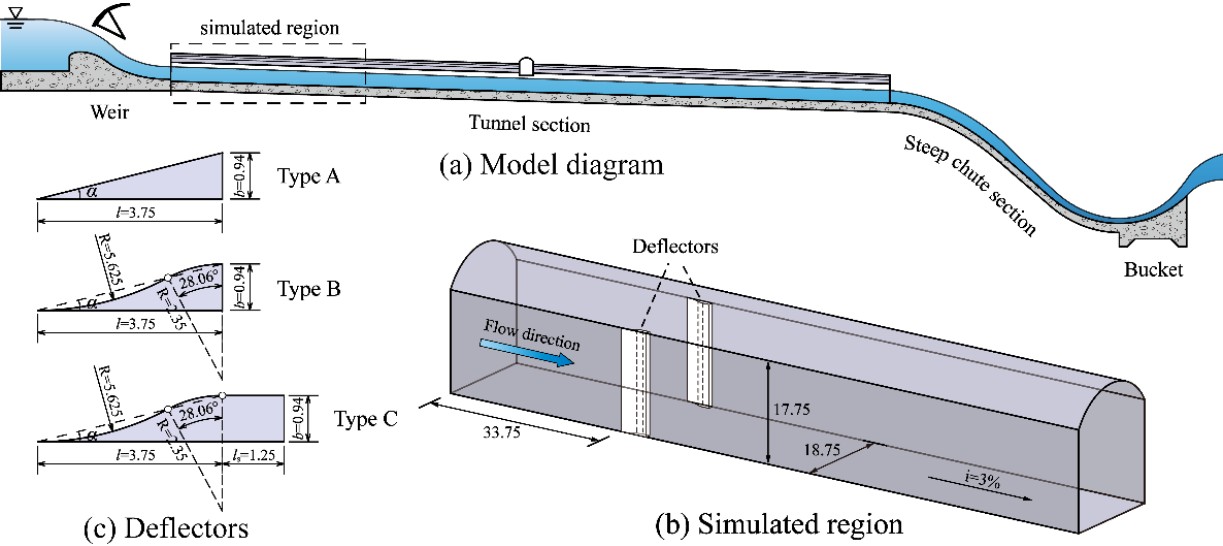

**Figure 1.** (**a**) Schematic diagram of the experimental spillway tunnel and details of (**b**) the simulated region and (**c**) the three lateral deflectors (unit is cm).

However, as the desired maximum operation head further increased, the flow velocity in the gentle-slope tunnel could exceed the threshold value of some 25 m/s [8], which consequently increases the cavitation risk of the tunnel [5,7,12] and the energy dissipation burden of the downstream chute. Under such conditions, aerators have to be installed to reduce cavitation risk [9,13,14], and restrictions resulting from flow pattern control have to be adequately considered. The main challenges of designing bottom aerators for gentle-slope tunnels are the low aeration efficiency caused by cavity blockage [6,12,15,16] and the cavitation protection of tunnel sidewalls [17,18]. Moreover, Hager discovered that shock waves could also be introduced by local changes in the bottom profile [14], such as offset aerators and bottom deflectors [16,19]. Therefore, to improve the aeration efficiency, lateral deflectors are usually installed together with the bottom aerators [15,17,18,20–23], of which a typical application is the fall-expansion aerators that constructed at the inlet of the free-surface tunnel behind the gate chamber [1]. It has been discovered that the lateral deflectors could not only serve as effective aerators [17,18,24] independent of the tunnel slope, but also help dissipate a certain amount of kinetic energy [24–26] and reduce the energy dissipation pressure of downstream structures. From this point, the hydraulic characteristics of the standalone lateral deflector (SLD, i.e., without bottom aerators) deserve further investigation and it is expected that the SLD could achieve decent hydraulic performance with regard to flow pattern and aeration.

To date, the commonest lateral deflector design is the triangular deflector whose up- and downstream endpoints are connected to the tunnel sidewall abruptly, and without vertical variation. The most noteworthy two of the few exceptions are probably the right-angled tetrahedrons and vertical plate deflectors investigated by Hager [26]. With such designs, water-wings and shock waves are usually induced by the lateral deflectors and thus introducing additional denudation risks. For example, Nie et al. (2006) discovered the jet formed behind the deflector impacts the sidewall and then is deflected upstream, resulting in partial blockage of the lateral cavity and the unstable swirling flow inside the lateral cavity plays a vital role in causing surface erosion [27]. The water-wings [18,20,23,24,28–30] formation was observed downstream of the impact region, which further rises up to the tunnel soffit [31] and could possibly develop into air pocket

flow [16]. Shock waves [14,16,20,24,26,29,32] were also found to be induced by lateral deflection, which could propagate far downstream, mainly in a diamond shape with water crowns colliding alternatively in the middle and on the sidewalls; thereby, causing sidewall erosion with repeating impact and pressure fluctuation. Under highly aerated situations, shockwaves could cause overtopping or trigger conduits choke [14], which dramatically impede the air transportation and thereby harm for the aeration protection [23]. Existing studies mostly focus on the optimization of the size [21,24,26,28,30] and layout [1,26,29–31] of the traditional triangular deflector, and few attempts can be found to improve the flow pattern and energy dissipation by means of modifying the deflector geometries.

In this paper, two new deflector geometries are proposed for the sake of both flow control and energy dissipation. The hydraulic characteristics of the newly proposed deflectors and the traditional triangular deflector are comparatively investigated using the hybrid approach of model test and numerical simulation. Special focus is put on the flow pattern improvement. The energy dissipation characteristics of these three deflectors are also compared based on the simulation results.

## 2. Experiment Setup

The experiments were carried out with the physical model of a sagging dragon tail tunnel constructed at the State Key Laboratory of Water Resources and Hydropower Engineering Science, Wuhan University. In the experiment, water was supplied by a circulating system composed of an underground reservoir, pumps, and pipelines. The non-pressured section of the tunnel was a $B = 18.75$ cm wide, $h = 17.75$ cm deep flume inclined with a $i = 3\%$ bottom slope. The corresponding scale fell into the range of $1/24{\sim}1/80$ with regard to the typical width being $4.5{\sim}15$ m of the high unit-width-discharge free-surface tunnels (e.g., Yele, Xiaowan, Xiluodu) in China [8]. Experimental studies adopting similar geometric scale or model tunnel width can be found in [33–36]. The scale effect is considered acceptable as the main focus of this paper are the flow pattern and other macroscopical flow characteristics such as energy dissipation behaviors [36,37], which are much less sensitive to the model scale compared to two-phase flow characteristics such as air concentration, bubble size and air vent discharge [19,23]. The lateral deflectors were installed symmetrically (i.e., with identical geometry and streamwise location) on the sidewalls with our bottom aerators. The flume and deflectors are both made of plexiglass to provide a transparent view of the flow pattern. Three types of deflectors (Figure 1c) were investigated, the width $b$ of which were all 0.94 cm and the streamwise length of deflectors A and B was 3.75 cm, while deflector C was further extended 1.25 cm downstream with a straight guiding line parallel to the side walls. The detailed geometries of the three types of deflectors are sketched in Figure 1. Type A is the traditional triangular deflector, deflector B is composed of two arcs tangent to each other: one being negative and has a radius of 5.625 cm and the other being positive and features a radius of 2.35 cm. As for deflector C, it is the same as deflector B except for the aforementioned additional straight line. Thus, the deflectors feature an identical contraction ratio of 10% ($2b/B$) and a tangent value of $1/4$, and the dimensionless tail extension of deflector C with regard to the streamwise length of the curved section l is $1/3$.

In the experiment, the inflow conditions were controlled at the cross-section 33.75 cm upstream of the deflector. All the measurements in this paper were conducted under the situation of inflow depth $h_{in} = 11.25$ cm and volume flow rate $Q_{in} = 57.65$ L/s, featuring a Reynolds number $Re = \frac{Q_{in}}{h_{in}Rv} = 1.2 \times 10^5$ ($R$ represents the hydraulic diameter calculated as $\frac{bh_{in}}{b+2h_{in}}$ and $v = 1 \times 10^{-6}$ m$^2$/s is the kinematic viscosity of water). This $Re$ value being larger than $1 \times 10^5$ indicates the scale effect arising from viscous stress can be neglected according to [38]. The corresponding depth-width ratio of 0.6 in the experiment is a representative value for the real-world high-discharge flood tunnels [15,22,23,33–36] in China. In this study, the flow depth was measured using a fluviograph (accuracy $\pm$ 0.18 mm). The water discharge was monitored using an electromagnetic flowmeter (IFM4080K, Jiangsu Runyi Instrument Co., Ltd., Huaian, China), featuring accuracy of 0.1 L/s. The flow velocity

was measured with a propeller-type flow meter (LS300-A, Fuzhou Lesida Information Technology Co., Ltd., Fuzhou, China) featuring accuracy of 0.01 m/s.

## 3. Numerical Models and Simulation Setup

Numerical simulations were performed in this study to obtain full-field velocities and turbulence properties for the analysis of the underlying mechanisms of flow pattern improvement and energy dissipation characteristics. The simulations were conducted using the commercial software FLOW-3D, which is claimed to have advantages over other opponents for free-surface flows and has been widely used for spillway and tunnel flows [39–41].

### 3.1. Governing Equations

FLOW-3D utilizes the one-fluid framework for free-surface flow modeling. This was achieved by using the Tru-VOF [42] technique to dynamically track the interface and, in the meantime, impose proper boundary conditions at the free surface. In this way, the computational cost is significantly reduced. The continuity and momentum equations are:

$$\frac{\partial u_i}{\partial x_i} = 0 \tag{1}$$

$$\frac{\partial u_i}{\partial t} + u_j \frac{\partial u_i}{\partial x_j} = -\frac{1}{\rho}\frac{\partial p}{\partial x_i} + f_i + v\frac{\partial^2 u_i}{\partial x_j \partial x_j} - \frac{\partial}{\partial x_j}\left(\frac{2}{3}k\delta_{ij} - 2v_t S_{ij}\right) \tag{2}$$

Here, $\boldsymbol{u}$ is velocity, $p$ is the pressure, $\rho$ and $v$ are the fluid density and kinematic viscosity, respectively. $\boldsymbol{f}$ stands for the body force, $k$ and $\mu_t$ stand for the turbulent kinetic energy, and turbulent viscosity. $\delta_{ij}$ is the Kronecker delta, and $S_{ij}$ is the mean rate of strain tensor calculated as $\frac{1}{2}\left(\frac{\partial u_i}{\partial x_j} + \frac{\partial u_j}{\partial x_i}\right)$.

The VOF equation for interface tracking reads:

$$\frac{\partial \alpha}{\partial t} + u_i\frac{\partial \alpha}{\partial x_i} = 0 \tag{3}$$

where $\alpha$ is the volume fraction of the simulated fluid.

The RNG $k$-$\varepsilon$ turbulence model [43] was adopted to account for the turbulence contribution to the time-averaged momentum transport, in which a transport equation was solved for the turbulent kinetic energy $k$ and the turbulent dissipation rate $\varepsilon$, respectively:

$$\frac{\partial k}{\partial t} + \frac{\partial}{\partial x_i}(ku_i) = \frac{\partial}{\partial x_j}\left((v + \sigma_k v_t)\frac{\partial k}{\partial x_j}\right) + P_k - \varepsilon \tag{4}$$

$$\frac{\partial \varepsilon}{\partial t} + \frac{\partial}{\partial x_i}(\varepsilon u_i) = \frac{\partial}{\partial x_j}\left((v + \sigma_\varepsilon v_t)\frac{\partial \varepsilon}{\partial x_j}\right) + C_{1\varepsilon}P_k\frac{\varepsilon}{k} - C_{2\varepsilon}^*\rho\frac{\varepsilon^2}{k} \tag{5}$$

where $v_t = C_\mu\frac{k^2}{\varepsilon}$, $P_k = 2v_t S_{ij}S_{ij}$, $C_{2\varepsilon}^* = C_{2\varepsilon} + \frac{C_\mu \eta^3\left(1-\frac{\eta}{\eta_0}\right)}{1+\beta\eta^3}$, $\eta = \left(2S_{ij}\cdot S_{ij}\right)^{1/2}\frac{k}{\varepsilon}$, and model parameters are: $\sigma_k = \sigma_\varepsilon = 1.39$, $C_\mu = 0.085$, $C_{1\varepsilon} = 1.42$, $C_{2\varepsilon} = 1.68$, $\eta_0 = 4.38$, and $\beta = 0.012$.

### 3.2. Simulation Setup

The computational domain adopted the same coordinate system as the experiment and extends from $x = -33.75$ to $x = 135$ cm in the streamwise direction. A grid convergence study involving three mesh schemes was carried out in advance to select a proper mesh for the simulation. The details of the grid convergence study were shown in Appendix A. One structured mesh block was used to discretize the domain into 10.25 million cuboid cells, the majority of which feature an average size of 0.34 cm × 0.125 cm × 0.2 cm ($x \times y \times z$). To

ensure the spatial resolution of the deflector geometry using the FAVOR® technique [44,45], the computational domain was rotated by the flume slope of 3% to be aligned with the $x$-coordinate and the gravity vector $\boldsymbol{g}$ was adjusted accordingly. Moreover, the domain was locally refined to 0.188 cm × 0.094 cm × 0.188 cm in the region of $-7.5 < x < 30$ cm, $-10 < y < -6.25$ cm and $6.25 < y < -10$ cm, and $0 < z < 12.5$ cm.

The outlet boundary was configured with the outflow boundary condition (BC), implying a zero-gradient condition since the outflow is supercritical. The bottom and sidewalls were set as non-slip walls with equivalent roughness $k_s = 0.015$ mm according to the plexiglass surface properties. The top boundary is specified with fixed relative pressure $p = 0$ (i.e., atmosphere pressure). As for the inlet boundary, a pre-simulation of the entire physical model (i.e., from the reservoir to the downstream cushion pool) was conducted first, and then all the flow parameters at $x = -33.75$ cm were mapped onto the inlet of the short domain using the grid overlay BC. The illustrative diagram of the simulation setup is shown in Figure 2.

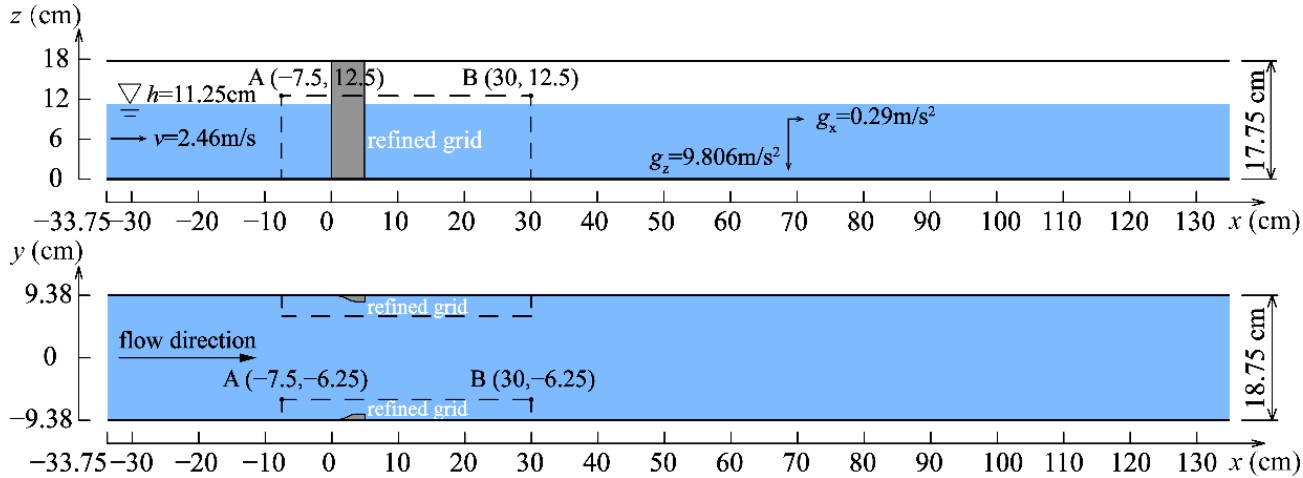

**Figure 2.** Schematic diagram of the simulation setup.

## 4. Results and Discussion

### 4.1. Model Validation

The simulated and measured water surface profiles at two longitudinal slices are comparatively shown in Figure 3a. It can be seen that the simulation results are in good agreement with the experimental data. Only small deviations can be found near the deflectors ($-5 < x < 10$ cm at $y = -8.375$ cm). Moreover, vertical and horizontal profiles of the measured and calculated streamwise velocity at $x = 112.5$ cm are presented in Figure 3b and c respectively. The simulated vertical velocity profile overlaps well with the measured data in the middle of the flume (i.e., $y = 0$ cm), whereas the calculated velocities near the sidewall ($y = -8.38$ cm) are approximately 10% smaller than the experimental data, which can be attributed to the wall function effects. As for the horizontal velocity distribution, the simulated velocities are also generally consistent with the measured data, with the bottom values ($z = 1$ cm) noticeably lower than those in the middle ($z = 6$ cm) and near the surface ($z = 11$ cm). While the bottom and middle velocities exhibit typical U-shape horizontal profiles, the surface velocities show a remarkable attenuation from the middle to both sides due to influence of shock waves, which are also reflected in the fluctuant measured data. Nevertheless, the overall accuracy of the simulation is considered acceptable to support the analysis of the flow pattern and energy dissipation characteristics.

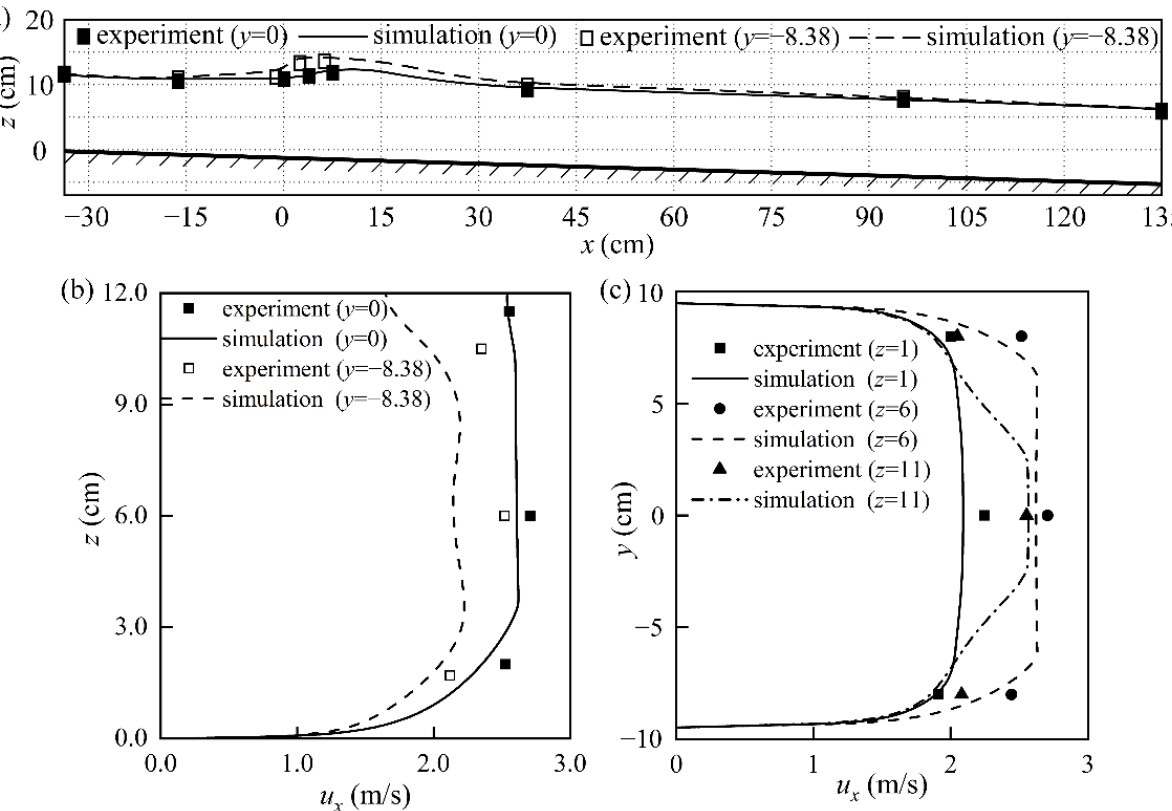

**Figure 3.** Comparison of the experimental and numerical results: (**a**) water surface profile; (**b**) vertical; and (**c**) horizontal distribution of the streamwise velocity $u_x$ at $x = 112.5$ cm. The unit of the coordinates is cm.

### 4.2. Flow Pattern

Figure 4 shows the side view of the flow pattern downstream of three different deflectors. Lateral jets can be found to form behind the deflectors, and the jets travel further at higher elevations, leading to a non-uniform lateral cavity that is consequently longer and wider at higher elevations. The jet impacting the sidewall is deflected immediately and keeps rising, clinging to the sidewall, associated with noticeable turbulence and aeration. In contrast, the water below the impacting region remains non-aerated. The three deflectors also form different boundary lines between the white and black water regions. With type A, the boundary line is smooth, above which the deflected flow develops into water-wings and continually rises over the sidewalls. With type B, the boundary line is fluctuant, and the deflected flow develops into water-wings that intermittently jump over the sidewall, and a few bubbles could be found in the lower clear water. As for type C, neither distinguished boundary line nor water-wings can be found since the aeration in the lower region is also noticeable. It appears that the deflected flow is submerged in the lateral jet and induces stronger turbulence and intensive aeration since a larger amount of air bubbles can be seen at the impacting region throughout the flow depth. This is probably caused by the rapid close of the lateral cavity and the stronger interaction between the jet and the deflected flow.

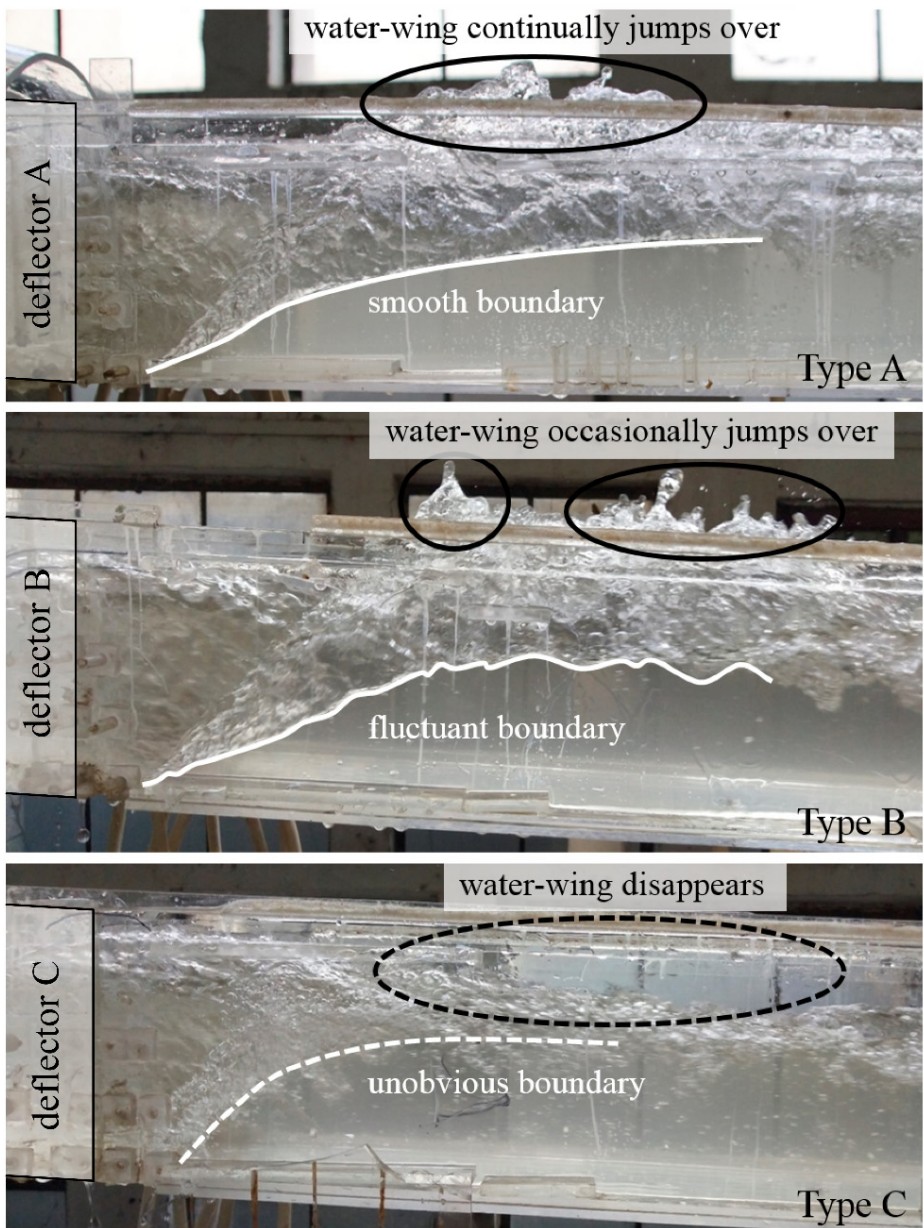

**Figure 4.** Side view of the flow pattern downstream of the deflectors.

The jet trajectories behind three different deflectors are marked with white lines and shown in Figure 5. The detaching jet behind deflector A moves in the direction tangent to the hypotenuse, leading to the widest and largest lateral cavity, which further leaves enough space for the lower deflected flow to develop to the upper region without contacting the jet trajectory. However, the jet behind deflector B exhibits a fluctuating phenomenon and forms an unstable cavity smaller and narrower than that behind deflector A. The fluctuating phenomenon can be mainly ascribed to the continuously varying edge slope of the deflector since it could yield non-uniform velocity distribution that exacerbates jet surface fluctuations. A benefit from the additional straight line in comparison to deflector B, the flow at the tail of deflector C has developed to a more uniform condition before detaching away from the deflector. It exhibits a trajectory almost parallel to the sidewall right behind the deflector, forming the narrowest and smallest lateral cavity. The rapidly closed cavity blocks the rising passage of the underneath deflected flow, preventing the formation of water-wings.

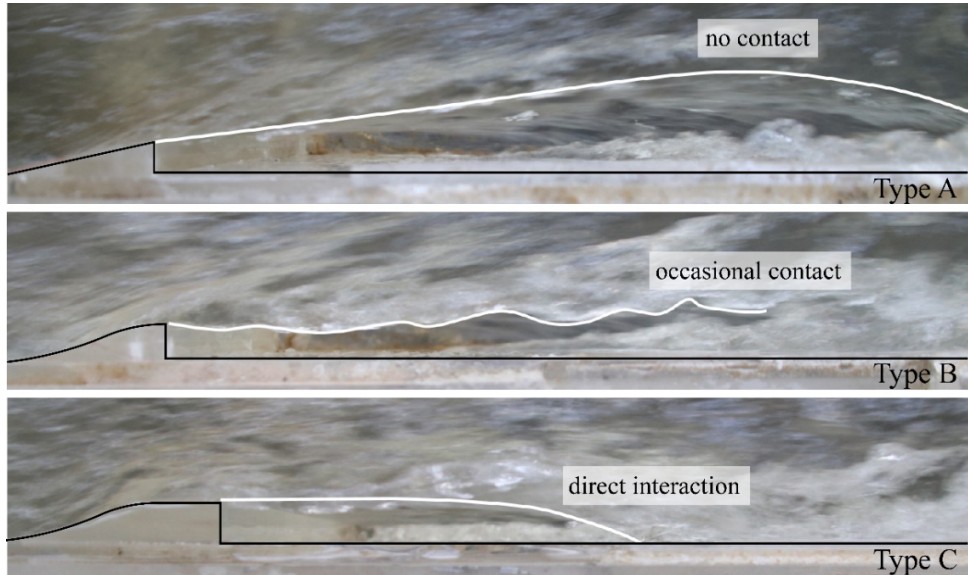

**Figure 5.** Top view of the flow pattern near the lateral cavity.

The development of water-wings and the subsequent shock waves downstream of the three deflectors are shown in Figure 6. It can be discovered that deflectors A and B form shock waves in bundle shape and diamond shape, respectively, whereas no shock wave is observed downstream of deflector C. With deflector A, the water-wings from both sides quickly rise up and detach from the main flow, jetting to the middle and colliding with each other in the air, and then fall back to the main flow with a certain vertical velocity and the lateral momentum dissipated. The returned water further produces disturbance on the surface and thereby inducing a bundle-shaped shock wave. With deflector B, the lateral momentum of the water-wings partially counteracts the jet in the cavity, and thus the water-wings fail to contact each other before falling back to the main flow. The returned water still contains certain lateral momentum and moves together with the main flow, producing two repeated reverse oblique developing lines on the flow surface and finally developing into a diamond-shaped shock wave. As for deflector C, the deflected flow is restricted in the relatively small cavity and thus only forms small water crowns, which are restricted in the vicinity of the sidewall and disappear further downstream.

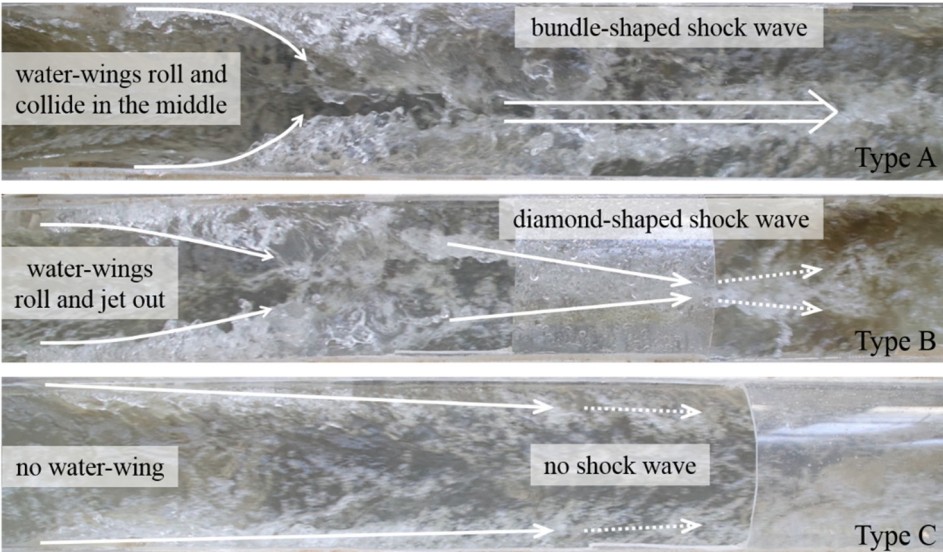

**Figure 6.** Top view of the flow pattern downstream of the deflectors.

Based on the above observations, it can be concluded that a continuous variation of the lateral deflector surface at the tail with an additional flow guiding extension is the key to the elimination of the water-wings and shock waves.

### 4.3. Velocity Distribution

The near-bottom ($z$ = 1 cm) velocity distribution around three deflectors is comparatively shown in Figure 7. The jets behind deflectors A and B are firstly contracted and reach the maximum cavity width at approximately 1/3 of the cavity length. After that, the jets restart to spread to the sidewall. In comparison, the jet behind deflector C continuously moves close to the sidewall. Consequently, deflector A exhibits the largest cavity size, followed by deflectors B and C, which is in agreement with the experimental observation. It is worth emphasizing that the velocity vector distribution around deflector B is apparently more non-uniform compared to those around deflectors A and C, which is the reason for the fluctuating phenomenon of the jet surface. Moreover, the maximum velocity behind deflector C is about 0.2 m/s lower than those behind the other two deflectors, and the near-wall low-velocity region (i.e., the purple dashed rectangle in Figure 7) behind deflector C is obviously larger than those behind deflectors A and B. These phenomena all imply more effective energy dissipation of deflector C compared to A and B.

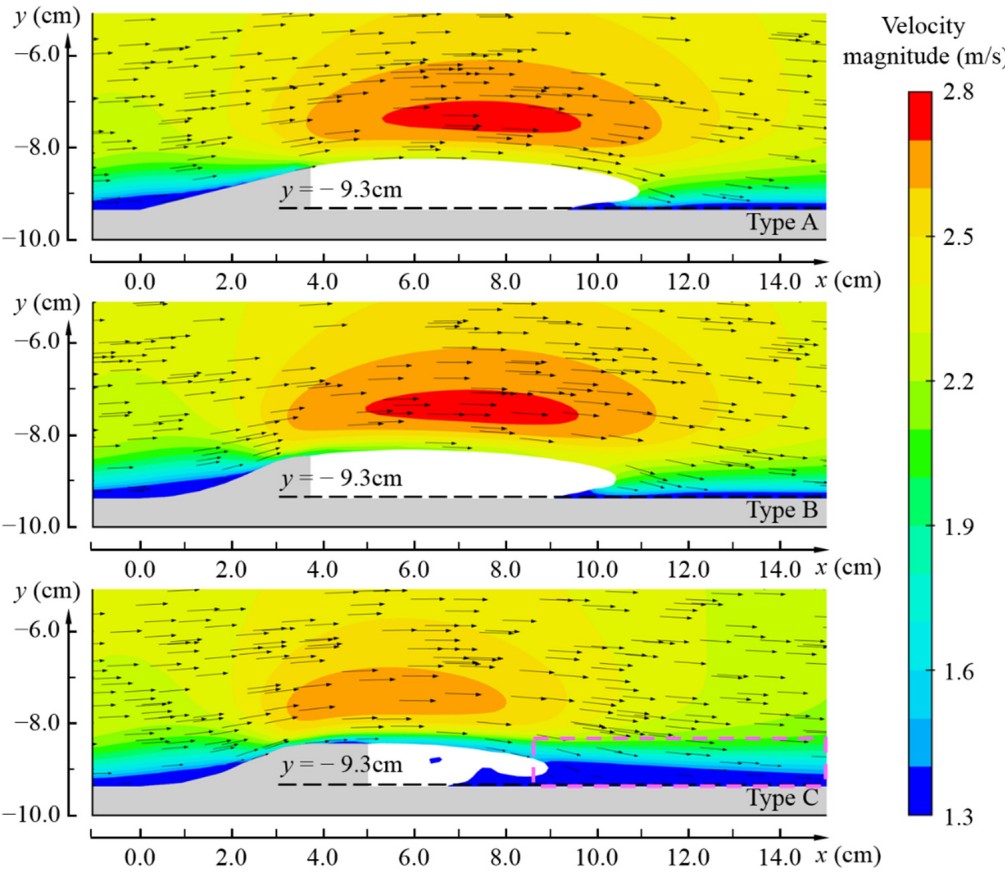

**Figure 7.** Horizontal velocity distribution around the deflectors at $z$ = 1.0 cm. The vectors are generated using the same density and scale for the three types of deflectors.

The near-wall ($y$ = −9.3 cm) vertical distribution of velocity behind the deflectors is comparatively shown in Figure 8 to illustrate the kinematic characteristics of the deflected flow. Distinguished by the velocity magnitude and vector direction, the longitudinal flow behind deflectors A and B can be divided into three distinct regions: the main flow region that flows downstream, the water-wing region that rises up while flowing downstream and the impacting region that exhibits the highest velocities and separating the above two

regions. The rising water-wing regions are in agreement with the experimentally observed boundary lines. As for deflector C, the flow direction is almost not affected by the jet since only a small region can be found that has upward velocities and therefore is the underlying reason for the fast recovery of the rising surface and the elimination of shock waves. Moreover, the velocities behind deflector C are lower than those behind deflectors A and B, especially dropping about 50% at the impacting region. This again confirms the more effective energy dissipation of deflector C.

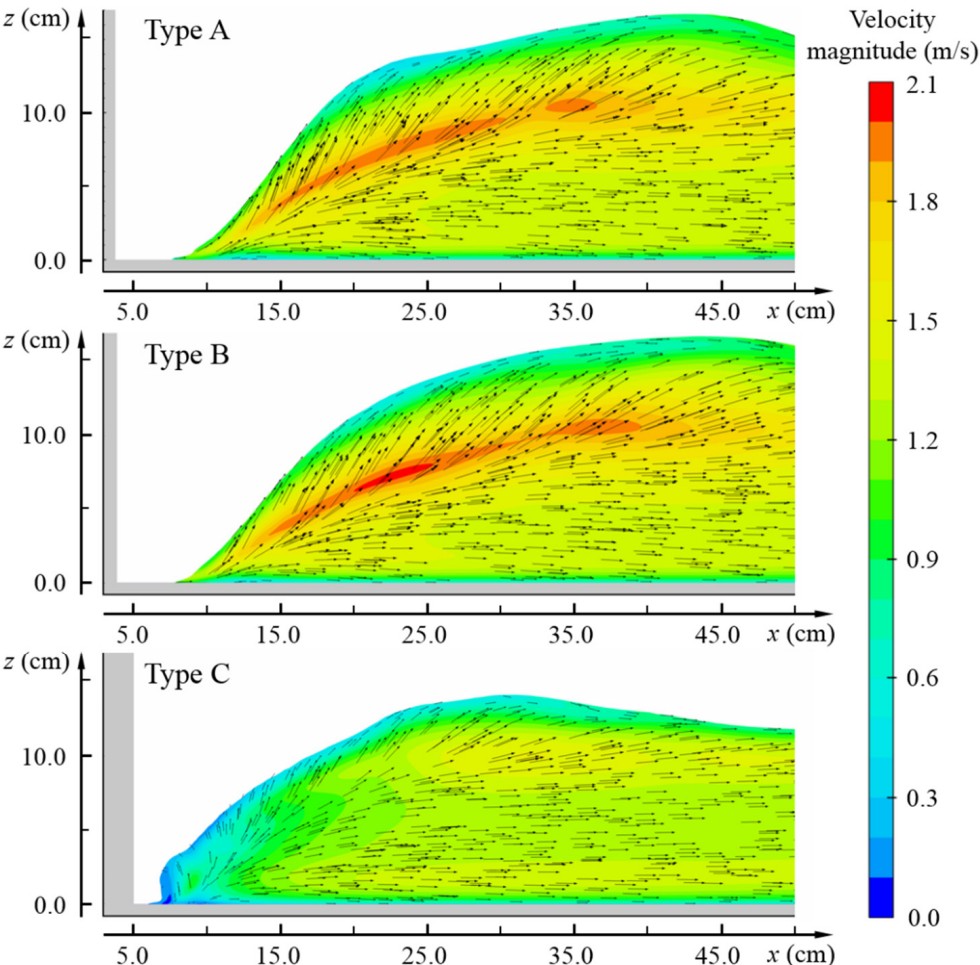

**Figure 8.** Velocity distribution near the sidewall ($y = -9.3$ cm). The vectors are generated using the same density and scale for the three types of deflectors.

Figure 9 shows the streamwise development of the cross-sectional velocity (CSV) with three different and without deflectors, which is computed from 17 flux surfaces with a spacing of 7.5 cm. The no-deflector scheme shows a linearly increased CSV, whereas for the other three schemes, the CSVs increase at the deflector region and decrease at the cavity region because of lateral flow contractions and expansions. With deflectors A and B, the CSVs first increase in the front of the cavity region and then decrease, whereas the CSVs of the flow behind deflector C exhibits a consistent decreasing trend due to continuous flow expansion. Affected by the shock waves, the CSVs of the downstream flow with deflectors A and B exhibit some fluctuations, and the fluctuating amplitude of deflector B is higher than that of deflector A. In contrast, the CSVs of deflector C feature a linear increase behavior similar to that of the no-deflector scheme in this region, although with lower values. Moreover, it can be found that the efficient energy dissipation mainly occurs at the impacting region (i.e., approximately $15 < x < 40$ cm).

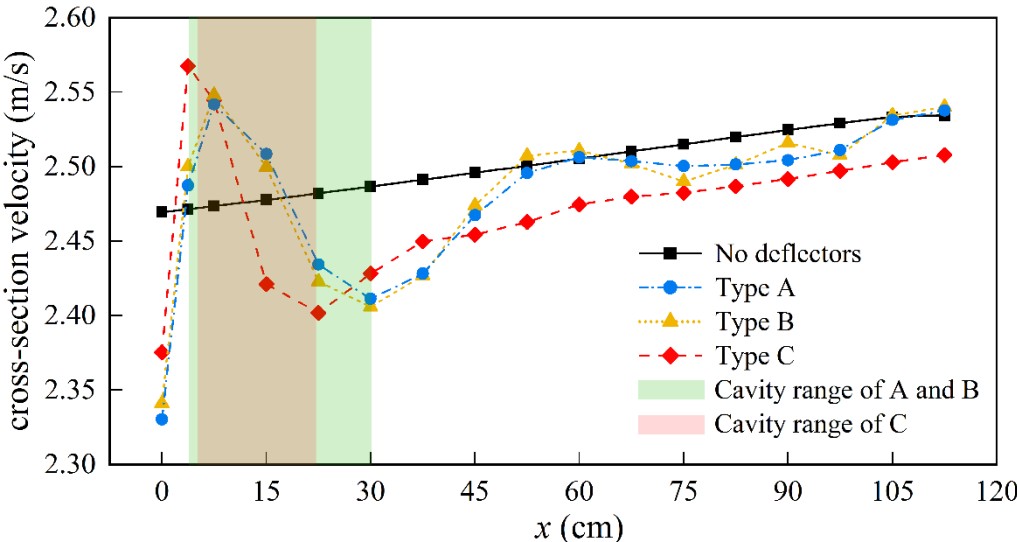

**Figure 9.** Streamwise development of the cross-sectional velocity along the flume with three different deflectors and without deflectors. The brown color indicates regions where the cavity range of C (pink) overlaps those of A and B (green).

### 4.4. Energy Dissipation Characteristics

Figure 10 shows the streamwise development of the flux-averaged hydraulic head (FAHH) along the flume with three different deflectors and without deflectors. In FLOW-3D, the FAHH is recommended to evaluate the hydraulic head where significant changes occur. It is calculated using the following equation:

$$h = \frac{\int \left( \frac{u^2}{2g} + \frac{p}{\rho g} + z \right) ds}{\int \varphi ds} \tag{6}$$

here, $h$ is the FAHH, $\varphi$ is the flux across a cell surface, and $ds$ is the open area of the cell face.

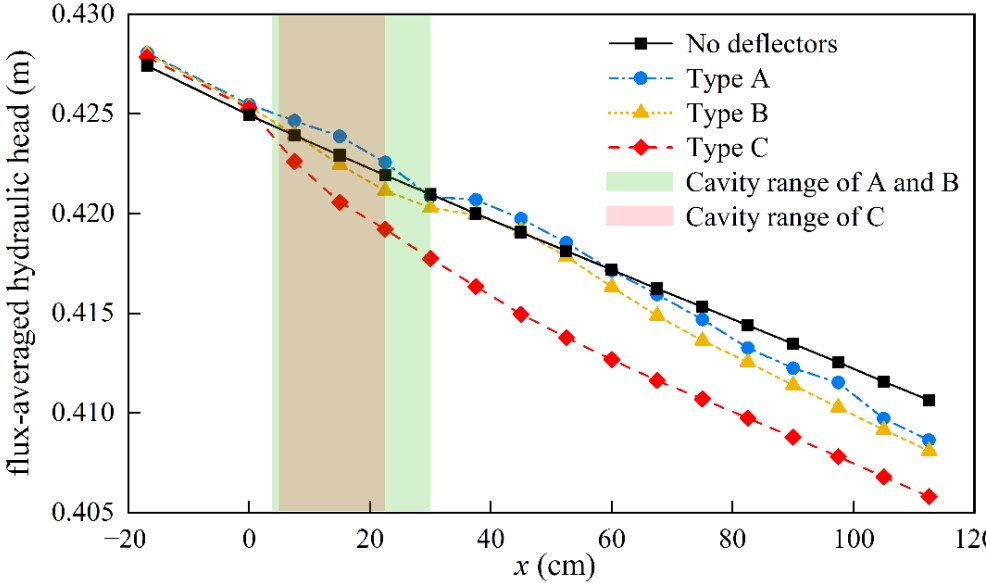

**Figure 10.** Streamwise development of the flux-averaged hydraulic head (FAHH) along the flume with three different and without deflectors. The brown color indicates regions where the cavity range of C (pink) overlaps those of A and B (green).

The FAHH decreases linearly in the flume without lateral deflectors. The FAHH of the flow underwent some certain fluctuations around the no-deflector values when passing deflectors A and B. These fluctuations continue until they are twice the cavity length downstream (i.e., to approximately $x = 60$ cm). Further downstream, the FAHH values of the flow behind deflectors A and B turn out to be lower than the no-deflector values. In contrast, a rapid decrease in FAHH can be found at the deflector and cavity region ($0 < x < 20$) for the flume equipped with deflector C, and further downstream ($x \geq 45$), the values of FAHH decrease steadily with a similar rate as those of the no-deflector scheme. A quantitative comparison of the energy dissipation effect of the three deflectors can be achieved using the local head loss coefficient $\zeta = (\text{FAHH}_1 - \text{FAHH}_2) \cdot 2g/v^2$ between $x = -17$ cm and $x = 45$ cm with $v$ being a reference CSV calculated at $x = -17$ cm. The local head loss coefficients for deflectors A, B and C are 2.72%, 2.90% and 4.2%, respectively. This behavior is in agreement with the streamwise development of the CSV and confirms the kinetic energy is mainly dissipated at the deflector and jet region.

To further analyze the energy dissipation mechanism of the lateral deflectors, the distribution of turbulent kinetic energy $k$ and turbulence dissipation rate $\varepsilon$ in the deflector and jet region are comparatively shown in Figure 11. The most noticeable difference in Figure 11 is the larger $k$ and $\varepsilon$ values at the impacting region behind deflector C compared to those behind deflectors A and B. This implies more mean kinetic energies are turned into turbulent energies and are then dissipated for the flow behind deflector C. This is consistent with the intensive aeration observed in the flow photos shown in Figure 6, as higher turbulent levels induce more intensive aeration [44–46].

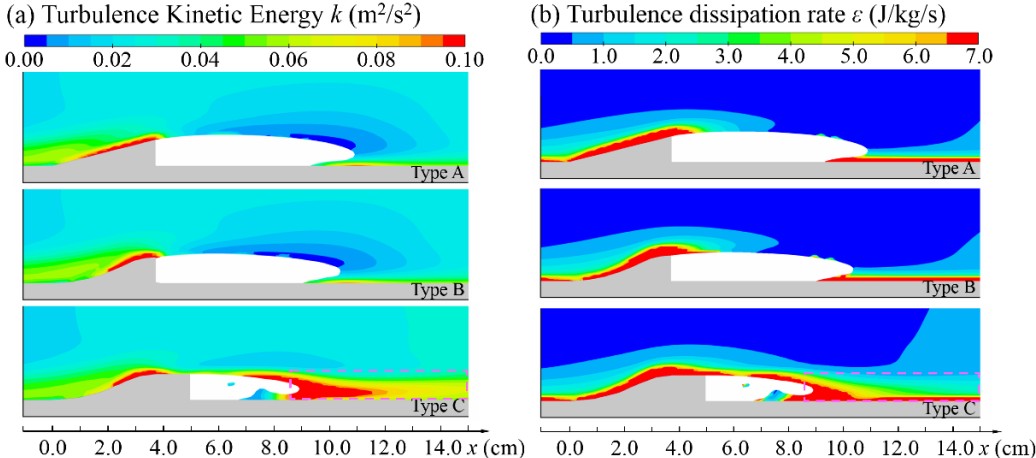

**Figure 11.** Horizontal distribution of (**a**) turbulent kinetic energy $k$; (**b**) turbulence dissipation rate $\varepsilon$ in deflector and cavity region at $z = 1$ cm.

## 5. Conclusions

In this paper, the hydraulic characteristics of lateral deflectors with three different geometries in gentle-slope free-surface tunnels were investigated using the hybrid approach of model test and numerical simulation. Special focus was placed on the flow pattern and energy dissipation features. The main findings are:

1.  The cavity formed behind lateral deflectors usually features a right-angled trapezoid shape with a larger streamwise length at higher elevations because of non-uniform velocity distributions. This makes the deflected flow rise up along the impacting region inside the cavity and potentially induced shock waves depending on the interaction of the rising up water-wings and the jet surfaces.

2.  The traditional triangular deflector forms an adequately wide cavity that allows for the free rising up of the water-wings inside the cavity, which further contributes to the development of the buddle-type shock wave, whereas the two-arc deflector yields a jet with fluctuating surface, resulting from the non-uniform planar velocity distribution

caused by the continuously varying curvature of the arcs. Water-wings also develop inside the cavity and eventually produce a diamond-type shock wave downstream. In contrast, the jet behind the two-arc deflector with a straight guiding line at the tail is stabler and travels a shorter distance before impacting the side wall. The jet could thus restrict the development of the rising flow, and thereby eliminate the formation of water-wings and shock waves. Based on these observations, it is concluded that a continuous variation of the lateral deflector surface at the tail with an additional flow guiding extension is the key to the elimination of the water-wings and shock waves.

3. Compared to the triangular deflector and the two-arc deflector, the two-arc deflector with a straight line exhibits more effective energy dissipation, as reflected in the local energy loss coefficient. The underlying reason for its effective energy dissipation is the more intensive turbulence introduced by the stronger interaction between the deflected flow and the jet surface, which also leads to more intensive aeration.

Compared to real-world engineering problems, the present study has some limitations. The physical model is roughly 1/24~1/80 of real-world free-surface tunnels, and thus scale effect is expected, in particular for air–water two-phase flow characteristics [47–49]. However, the scale of the physical model is considered acceptable with regards to the main concern of the current paper, which is the comparative evaluation of the different lateral deflectors in terms of the flow pattern and the energy dissipation behaviors. Particularly, the jet trajectory behind the lateral deflector and its interaction with the deflected flow could be decently reproduced in the physical model. Moreover, considering that the numerical simulation approach remains immature for highly aerated flows but much more reliable for non-aerated flow [13,50,51], the hybrid approaches used in this paper could offer valuable insight into the underlying reason for the flow pattern improvement and higher energy dissipation observed with deflector type C. Aeration is another key concern for lateral deflector designs but is not investigated in detail in this paper due to facility reasons. Nonetheless, the findings from the study provide a novel and preferable lateral deflector design for gentle-slope free-surface tunnels, which could, if not resolve, significantly improve the unwanted flow patterns of water-wings and shock waves. Meanwhile, it could also achieve improved energy dissipation compared to traditional alternatives. The aeration characteristics of the novel deflector will be investigated in future research.

**Author Contributions:** The conceptualization of this research was directed by J.W. and the experimental data were from J.D. and S.D.; the numerical modeling was performed by J.D. under the supervision of J.W. and Z.D.; the validation is carried by J.D. and S.D.; the manuscript was mainly finished by J.D., Z.D. and J.W. All authors have read and agreed to the published version of the manuscript.

**Funding:** This research was funded by the National Key Research and Development Program of China (Grant No. 2021YFC3090105), the China Postdoctoral Science Foundation (No. 2021M692754), and Guizhou Science and Technology Joint Support (2019) No. 2890.

**Institutional Review Board Statement:** Not applicable.

**Informed Consent Statement:** Not applicable.

**Data Availability Statement:** Not applicable.

**Conflicts of Interest:** The authors declare no conflict of interest.

**Appendix A**

A grid convergence study involving three mesh schemes was carried out to evaluate the grid quality. The cell size ($x \times y \times z$) in the refined region of the three mesh schemes was 0.625 cm $\times$ 0.313 cm $\times$ 0.625 cm (coarse), 0.375 cm $\times$ 0.188 cm $\times$ 0.375 cm (medium), and 0.188 cm $\times$ 0.094 cm $\times$ 0.188 cm (fine), respectively.

The grid spacing of the 3D grids is defined as $h_k = \sqrt[3]{h_{x,k} \cdot h_{y,k} \cdot h_{z,k}}$. The computational solution $f$ for a grid $k$ is defined as the computed maximum streamwise velocities $u_{\max}$ in the middle longitudinal section at $x$ = 112.5 cm. The *GCI* for each grid $k$ is computed

as $GCI_k = F_s|E_k|$, where a value of 1.25 is chosen for the factor of safety $F_s$, following the recommendations of Roache [52].

Table A1 shows the grid convergence index (GCI) calculated using the approach of Roache and Salas [52]. The observed order of accuracy is $p = 1.84$, and the *GCI* shows a convergence trend. The fine grid takes a very satisfactory value of 0.224%, which is an order of magnitude higher than the former and therefore is considered fine enough to resolve the flow field.

**Table A1.** Parameter of grid convergence calculation.

| Mesh | $h$/cm | $r$ | $p(u_{max})$ | $f(u_{max})$/(m·s$^{-1}$) | $\varepsilon(u_{max})$/% | $GCI(u_{max})$/% |
|---|---|---|---|---|---|---|
| coarse | 0.496 | - | | 2.607 | - | - |
| medium | 0.297 | 1.67 | 1.84 | 2.633 | 0.997 | 2.040 |
| fine | 0.149 | 1.99 | | 2.621 | 0.456 | 0.224 |

The vertical distributions of streamwise velocity $u_x$ and turbulent kinetic energy $k$ in the middle longitudinal section at $x = 112.5$ cm are comparatively shown in Figure A1. From Figure A1, it can be found that the main deviations of the $u_x$ profiles lie in the region close to the water surface and the bottom wall, whereas in the middle, all three meshes return almost identical velocity values. Moreover, the difference between the medium and fine meshes is less obvious compared to that between the coarse and the medium meshes. As for the $k$ profile, the result from the coarse mesh is strikingly different from those from the medium and fine meshes, the difference between which is relatively negligible. Therefore, the fine grid was selected for the simulation.

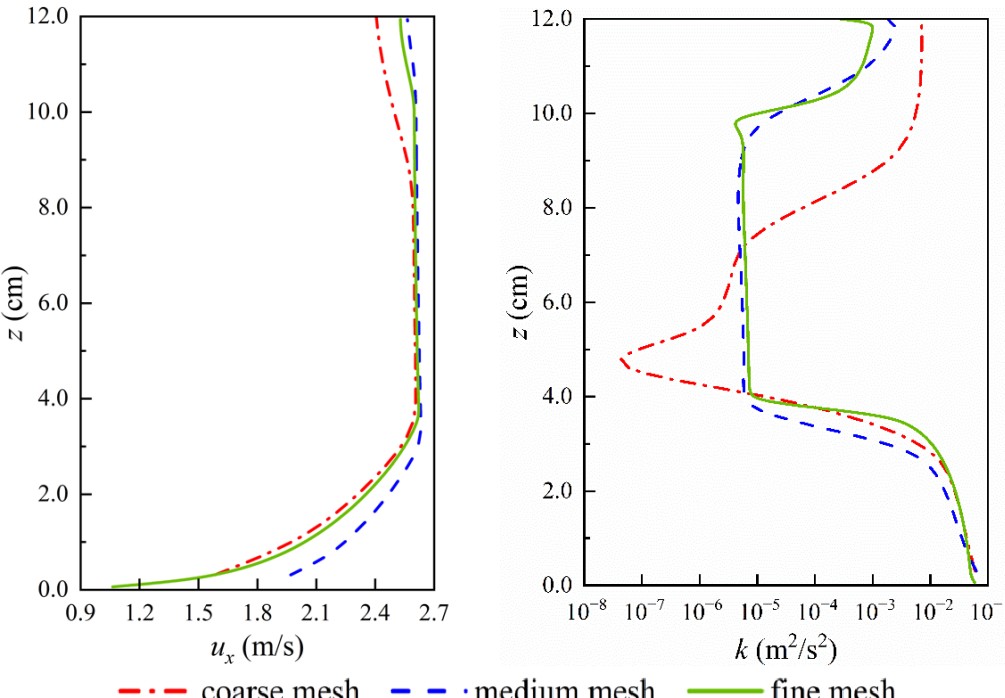

**Figure A1.** Vertical distributions of streamwise velocity $u_x$ and turbulent kinetic energy $k$ in the middle longitudinal section at $x = 112.5$ cm.

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
