# Peer review of "Hydraulic Characteristics of Lateral Deflectors with Different Geometries in Gentle-Slope Free-Surface Tunnels"

_water, doi:10.3390/w14172689_

Round 1

Reviewer 1 Report

The authors presented an investigation on the lateral deflectors in free surface tunnel. Three geometries of deflector structures were studied, including a traditional triangle style and two proposed designs. The flow pattern and cavity range were discussed both in experimental test and numerical simulation. In all, the manuscript is well organized and well written. However, there are still few points should be supplemented or clarified before commendation of publishing.

1.      It seems that Type B and Type A perform similar while Type C is remarkably improved both on flow pattern and cavity range. For a direct comparison, a Type D with a triangle start and an extension like Type C should be carried out too. It is necessary to claim whether the deformation of triangle shape start or the extension is the root reason that affect all the performances.

2.      In figure 1, it seems that the inlet of this gate shape tunnel actually locates below the water level in the upstream reservoir. If so, the flow condition in the tube should be separated into a pressurized part (fulfilled the tube) and a free surface part, relative ref: ‘Improvement on Simulation Methods of Fluid Transient Processes in Turbine Tailrace Tunnel’. It has to be claimed why the authors only consider the inlet area as free surface in the very beginning.

3.      Series No.: There are two Eq.(3). Pls check the spelling errors throughout.

In summary, I would suggest a minor revision before acceptance.

Reviewer 2 Report

Please see the file attached.

Reviewer 3 Report

Major comment: Introduction and conclusions should be improved. 

Potential threats of occurrence of diamond-shaped shockwaves and bundle-shaped shock waves in the tunnel with regard to free surface flow or pressurized flow should be explained better.

Minor comments:

1. Deflectors are not shown in Fig. 4

2. Spelling mistakes should be removed thoroughly for example 'Wier'

3. Discussion (section 5) should be changed to Conclusions

Round 2

Reviewer 2 Report

The Authors addressed my concerns with a special attention. I'm very satisfied with this revised manuscript and I would recommend to accept it in present form.